# Relationship between Consumers’ Perceptions about Goat Kid Meat and Meat Sensory Appraisal

**DOI:** 10.3390/ani13142383

**Published:** 2023-07-22

**Authors:** María J. Alcalde, Guillermo Ripoll, María M. Campo, Alberto Horcada, Begoña Panea

**Affiliations:** 1Agronomy Department, University of Seville, 41013 Seville, Spain; albertohi@us.es; 2Animal Science Department, Centro de Investigación y Tecnología Agroalimentaria de Aragón (CITA), Avda. Montañana 930, 50059 Zaragoza, Spain; gripoll@cita-aragon.es (G.R.); bpanea@cita-aragon.es (B.P.); 3Instituto Agroalimentario de Aragón—IA2 (CITA-Universidad de Zaragoza), Avda. Miguel Servet, 177, 50013 Zaragoza, Spain; 4Department of Animal Production and Food Technology, University of Zaragoza, C/Miguel Servet, 177, 50013 Zaragoza, Spain; marimar@unizar.es

**Keywords:** goat kid meat, survey, home test, consumer clusters

## Abstract

**Simple Summary:**

The type of goat kid meat that is consumed in countries of the Mediterranean basin belongs to very young animals with little fat. Despite being a healthy meat from a nutritional point of view, its consumption is not very common. In this study, after conducting surveys with a number of different families, we evaluated goat kid meat using a home test sensory analysis. Despite the meat obtaining high scores during sensory analysis, there is often considerable ignorance about the factors that affect the production of this type of animal, especially in young people and in areas where consumption of this meat is uncommon. For this reason, it is important to show its value and to improve production and profitability in the primary sector, carrying out specific marketing work to target the different clusters obtained in the study.

**Abstract:**

The goat meat preferred by consumers in Spain comes from suckling goat kids, slaughtered at a live weight of 8–10 kg. However, consumption of this meat is very uncommon, so it is necessary to show its value. To achieve this, we planned to investigate consumers’ perceptions about goat kid meat and to study whether their perceptions are related to their sensory appraisal of the meat, measured by the mean of the consumers’ home tests. The experiment was conducted with 30 volunteer families (from two cities with different consumption patterns), who were surveyed regarding their sociodemographic parameters, purchasing and eating habits, and the importance of the meat’s attributes. As a result, four clusters were obtained, which were labeled “negative,” “idealistic,” “positive,” and “concerned about fat content”. The parameters of the animal production system were valued differently by the clusters. Meat tenderness, taste, and amount of fat were considered very decisive factors by most respondents. When the goat kid meat was valued, tenderness was considered more important than taste among older people (“negative” cluster), whereas there was not so much difference between the appraisal of all parameters for the other three clusters. We conclude that it is necessary to improve the information received by consumers about goat production systems and meat quality parameters. There is certainly potential for creating new markets, especially targeted toward young consumers and considering specific strategies for the different groups of consumers, depending on the region and habits of consumption.

## 1. Introduction

The 11.5 million goats existing in the EU-27 in 2021 (FAOSTAT, 2023) are unequally distributed across countries. Greece has 24.8% of the total number, while Spain ranks second, with 22.6%, ahead of Romania, France, and Italy. These five countries account for over 82% of the goats in the EU. Although the goat sector’s contribution to total livestock production is very low, goats are essential in large areas of Mediterranean Europe to maintain the natural landscape, particularly of farmland considered to have a high natural value, and to generate income that improves the financial livelihood of farmers in depressed areas [1].

In Spain, the goat meat preferred by consumers comes from suckling kids [2]. These animals are fed only on milk and are slaughtered at a live weight of 8–10 kg, aged 35–45 days. In 2021, the average weight of goat carcasses produced in Spain was 8.6 kg and 74.6% of the carcasses were classified as suckling kids [3]. Goat kid meat plays an important role on special occasions, especially Easter, Christmas, and those of non-Christian faiths, which has a huge effect on the seasonal patterns of production, prices, and external trade. Goat kid meat is appreciated by consumers in the southern EU due to its natural origin, flavor, and taste [4]. In addition, it is considered by consumers as healthy white meat because of its low content in fat [5], and it does not, therefore, suffer the negative connotations that red meat has in general [6,7].

According to the Ministry of Agriculture, Fisheries and Food of Spanish Government (MAPA, 2023), in the last decade, the consumption per capita of sheep and goat meat decreased by around 70%, from 5.6 kg in 2003 to 1.7 kg in 2021, with only 0.07 kg of this corresponding to goat meat. For this reason, it is essential that goat meat is revalued. To achieve this, we need to identify trends [8] and consumer behaviors and preferences, as well as determine consumer segments [9,10].

Food consumption patterns have changed over recent decades, not only due to socioeconomic and cultural trends, but also because of an increasing diversification in the grouping of consumers [10], as well as the tendency toward consuming more plant-based foods [11]. Thus, the study of consumer behavior is of the utmost importance in the marketing of products. Consumer behavior is affected by the multiple attributes of the product, and the relative importance of a given attribute is reflected in its appeal to the consumer. While beliefs are linked to consumers’ knowledge about the product, attitudes refer to the affective response to the product. As a result, consumer perceptions are divided into three main areas: Cognitive (knowledge), affective (attitudes and feelings), and conative (behavior, purchasing intent, and real consumption) [12]. According to Cotes-Torres et al. [13], the factors affecting consumption decisions can be classified into demographic and psychographic elements, with the former referring to all variables that identify the individual (age, gender, social class, etc.) and the latter to all perceptions or beliefs [14]. However, one of the main psychological factors affecting consumer behavior is perception. Consumers associate a product group with a group of values on the basis of a set of cognitive categories and actions that define a lifestyle [15]. With regard to food, these lifestyles contain five interrelated elements: Forms of purchasing, factors used to evaluate the quality of foods, cooking methods, places where the food is consumed, and reasons for purchasing.

To investigate these food-related lifestyle factors, surveys are an extremely useful and widely used tool, since they allow to identify consumer segments [16,17,18,19]. Consumer segmentation improves our knowledge of the market and detects possible trends, pointing to marketing strategies that fit consumers’ requirements better than a marketing strategy designed for the average consumer [14].

There is a considerable body of literature concerning consumer preferences and segmentation. For example, Bernues et al. [20] reported four European consumer types according to the importance they placed on the extrinsic quality attributes of lamb, including animal feeding, the origin of meat or environmentally friendly production methods, the established relationships with purchasing motives, and the demand for information and sociodemographic aspects. Other studies have clustered consumers of lamb meat in different European countries according to several criteria, such as the type of feeding system, origin of the meat, and price [21,22] or according to their perceptions about pork meat, including intrinsic and extrinsic factors [23,24]. More recently, Miller [25], based on lamb meat, stated that flavor, diet, and animal age continue to be the main drivers of consumer preferences and concluded that lamb consumers’ profiles vary across countries, based on cultural effects and production practices.

Nevertheless, to the best of our knowledge, there is very little European literature regarding consumers’ preferences about goat kid meat [26,27,28,29], and practically no research in Europe has focused on consumption, the consumer’s perception of quality, and the market segmentation of goat kid meat consumption.

The main aim of this paper was to investigate consumers’ perceptions about goat kid meat and to study whether their perceptions are related to their sensory appraisal, measured by the mean of the consumers’ home tests.

## 2. Materials and Methods

### 2.1. Recruitment of Consumers

The experiment was conducted with 30 volunteer families, 15 from Zaragoza (northern Spain) and 15 from Seville (southern Spain). These two cities were chosen because they represent two areas in Spain with very different patterns of meat consumption, Zaragoza having well above the average and Seville well below the average meat consumption in Spain [30]. To be included, the families had to contain a minimum of three people over the age of 14 years.

### 2.2. Survey

The consumers had to answer a brief closed/open response questionnaire (Table 1), with different types of scales to measure the responses. It was divided into three parts:

The first part included open-ended questions aimed at collecting personal and family data (number and ages of family members, individual gender, and place of residence), as well as purchasing and eating habits (place of purchase, frequency of cooking, frequency of and place where goat kid meat is consumed, frequency of cooking habits, frequency of eating out, and frequency of eating pre-prepared meals). These responses were used to define the consumers’ profile.

The second part was designed to collect the consumers’ perceptions about goat kid meat, using a seven-point Likert scale. The results were used to categorize the consumers into clusters.

The third part of the questionnaire used a five-point Likert scale to collect information on the importance of several elements when consuming goat kid meat: Animal breed, age, live weight and feeding regime, origin of the meat, place of consumption, type of meat cut, amount of fat, meat color, tenderness, and taste.

After answering the survey, the consumers tasted the meat as part of a home test, as described below.

### 2.3. Consumer Home Test

For the tasting home test, we used the meat from a total of 210 male, single-birth suckling kids from seven Spanish goat breeds: Five meat breeds (Blanca Andaluza, Blanca Celtibérica, Moncaína, Negra Serrana-Castiza, and Pirenaica) and two dairy breeds (Malagueña and Murciano-Granadina). All goat kids were fed with maternal colostrum and maternal milk until slaughtered. The goats were grazed on native pastures and supplemented with commercial feed when the pasture was not able to cover their nutritional requirements. When the kids reached the target live weight, they were slaughtered using standard commercial procedures.

Detailed information about carcass characteristics and instrumental and sensory meat quality of the animals involved in this study can be seen in Panea et al. [5] and Ripoll et al. [31]. After 24 h of chilling following commercial protocols, both hind legs were removed from the carcasses [32]. Then, the legs were vacuum-packed, aged at 4 °C for three days, frozen, and kept at −18 °C until the home test was conducted.

Each family received one leg a week, codified with a three-digit number, for a total of 14 weeks. At the end of the trial, all of the families had tasted two legs of each of the seven breeds involved. To avoid confusion in the order or carry-over effects [33], the samples were supplied frozen to the families following a randomized design. The consumers had to defrost the legs at 4 °C overnight before consumption. For each family, it was mandatory to cook the legs in the same way and it was also recommended to roast them in a conventional oven without the addition of any spices, only salt. Next, each member of the family assessed the tenderness, flavor, juiciness, and overall appraisal of the meat using a structured scale from 0 (“dislike very much”) to 10 (“like very much”).

### 2.4. Data Analyses

First, we carried out a frequency distribution analysis of the sample population according to age, gender, and place of residence. Next, a hierarchical cluster analysis (using Ward’s method for aggregation and Euclidian distance) using the variables related to lifestyle and perception was performed to identify homogeneous groups of respondents according to their perceptions about goat kid meat. The number of clusters was obtained based on the R^2^ obtained and a strong increment produced in the Cubic Criterion of Clustering and Pseudo F values (XLstat v. 5.1.1410.0, 2023). Finally, to produce a profile of the resulting groups, a GLM test for continuous variables and a chi-square analysis for nominal variables were carried out. For the GLM procedure, differences between means were calculated with a Bonferroni test at 5% significance. In the χ^2^ test, we used a significant probability of less than 0.05. When more than 20% of the boxes had expected frequencies lower than five, the likelihood ratio was used at the same level of probability. When the adjusted standardized residuals between the observed and expected cases in each box were greater than |1.96|, we considered that there was a pattern of association between the studied variables.

## 3. Results and Discussion

### 3.1. Characterization of the Consumer Sample

A description of the sample is shown in Table 2. A total of 97 complete questionnaires were collected. The sample population surveyed was well balanced in terms of age, gender, and place of residence (Zaragoza vs. Sevilla).

### 3.2. Clustering and Sociodemographic Description

From the analysis of the consumers’ perceptions about goat kid meat, four clusters were created (Table 3 and Table 4). Cluster 1 was composed of 13.8% of the respondents, Cluster 2 of 34.0%, Cluster 3 of 27.7%, and Cluster 4 of 24.5%.

The clusters differed in the average age of the respondent, the age of the family members, and the place of residence (Table 3 and Table 4), but there was no difference in the gender of the respondents (*p* > 0.10). Thus, Cluster 1 was composed of older people, with an average age of 56 years, whereas the other clusters ranged from 36 to 44 years old, without any differences between them. Regarding family composition, there were slight differences in people under 14 years old, with less presence in Cluster 1 than in the others. People from Seville were mainly grouped in Cluster 4, whereas people from Zaragoza were mainly in Clusters 2 and 3.

The main differences among the clusters in terms of perceptions about goat kid meat (Table 5) concerned questions that compared kid and lamb meat (more expensive, healthier, containing more fat, or heavier) and the milking of goat kids.

Cluster 1 did not agree that goat kid meat is more expensive (*p* < 0.05), healthier (*p* < 0.05), or tastier (*p* < 0.05) than lamb meat. Furthermore, Cluster 1 stated that they did not prefer natural milking of the kids compared to artificial milking (*p* < 0.001), did not prefer a heavy goat kid over a light one (*p* < 0.001), and would not pay more for goat kid meat than lamb meat (*p* < 0.01). This cluster showed a neutral or negative perception of goat kid meat, so were labeled “negative”. The “negative” cluster was composed of older respondents (average 56 years old) and family units with less people under 14 years of age. This cluster was equally divided between both cities in the study.

The cluster of consumers with the most positive perception of goat kid meat was Cluster 3. Cluster 3 affirmed that kid meat is more expensive, healthier, and tastier than lamb meat (*p* < 0.05) and they did not think that kid meat contained more fat than lamb meat. Besides this, they did not prefer a goat kid to be milked naturally rather than artificially (*p* < 0.001). This cluster had a positive perception of goat kid meat, so were labeled “positive”.

Clusters 2 and 4 had intermediate perceptions, but with different preferences. Cluster 2 preferred goat kid meat that was fed via natural milking rather than artificial milking (*p* < 0.001) and thought that goat kid meat is tastier than lamb meat. This cluster did not have enough information to differentiate lamb from goat kid meat, but had an essential idea of what they wanted. They were therefore labeled “idealistic”. “Idealistic” consumers were younger than “negative” consumers (*p* < 0.05) and this was the cluster with more people under the age of 14 years at home and living in Zaragoza.

In the same way, Cluster 4 had no information about the differences between lamb and goat kid meat, but expressed the view that kid meat contains more fat than lamb meat (*p* < 0.01), and they also showed a strong preference for the meat from a heavy kid compared to a light kid (*p* < 0.001). Thus, this cluster was labeled “concerned with fat content”.

The “positive” cluster were middle-aged (average 44 years old), without differences in age compared to the “idealistic” and “concerned about fat content” clusters. They contained the same number of people under 14 years of age as the “concerned about fat content” cluster, and the main difference with this latter cluster was that most members of the “positive” cluster lived in Zaragoza, while most of the “concerned about fat content” cluster lived in Seville.

No differences were found between clusters in the intention to consume goat kid meat, and in all four clusters, the main reason was that they do not see it in the supermarket. Similarly, there were no differences whether it was of a quality brand or not, which disagrees with the conclusion by Cubero et al. [34], who stated that the PDO/PGI certification was always among the top three preferred attributes.

According to different studies [10,22,35], consumers distinguish lamb meat from other kinds of meat, among other reasons, for its higher price or its intense or unique taste. In the same way, Mandolesi et al. [36] concluded that consumers associate sheep and goat meat with a unique taste, authenticity, and natural production, linked with values such as health and enjoyment of life. In contrast, non-consumers feel disgusted when they think about eating these meats and do not associate any specific health benefits to their consumption, disliking their taste, odor, and fat content.

Due to the relationship between high-fat diets and heart disease, consumer interest in fat content and the fatty acid composition of foods has grown in recent years [37]. Nevertheless, our results show that consumers do not perceive goat kid meat as a fatty meat and, therefore, this does not seem to be a crucial factor regarding goat kid meat consumption.

On the other hand, the different diet (natural vs. artificial milk) is an aspect that has been considered in several studies; in these works, the acceptability of meat from suckling kids fed on natural milk was greater among older consumers and people with a moderate consumption of meat [28], which disagrees with the current results, since the cluster in our study that preferred natural milk was the “idealistic” cluster, with an average age of 38 years.

On the contrary, there is a general belief that, while goat meat is of inferior quality compared to mutton and beef, goat meat obtained from young, fattened animals is not inferior to lamb [38]. Of these, light suckling kids, such as those used in our study, are perceived by consumers to be high-quality meat [39].

Bernués et al. [10] found that place of residence, age, and level of formal education are more relevant for conveniently segmenting the lamb meat market than gender and income, whereas for Escriba-Perez et al. [19], the geographical area was one of the most important variables in relation to lamb meat consumption.

According to official statistics [40], the profile of sheep/goat meat-consuming Spanish households in 2020 corresponded to households made up of couples with older children or retired or adult couples without children. In 2021, the consumption of fresh goat kid meat in Aragon was 0.97 kg, while in Andalusia, consumption stood at 0.43 kg/per capita.

Consumption habits including diet and cultural factors are important in the assessment of overall acceptability, which is strongly influenced by consumers’ culinary background [41]. Furthermore, several studies have shown that food preferences developed in infancy and early childhood can have long-lasting effects on food preferences of individuals in later life [42].

For Rabadán et al. [43], the socioeconomic factors that most influenced preferences for lamb meat consumers were age and level of education. Wycherley et al. [44] found that young respondents could be categorized more frequently into “uninterested” groups, whereas middle- and old-aged respondents were more frequently found in “conservative (traditional)” groups. Ngomane et al. [45] suggested that there is a global trend toward an increase in the demand for goat meat, which is driven by the older generations, since goat meat does not appeal to young people.

### 3.3. Purchasing and Eating Habits

According to our results, 36% of the respondents bought meat in traditional butcher shops, 19% in markets, 39% in supermarkets, and 6% in hypermarkets. Corcoran et al. [46] showed that in Spain, most participants bought lamb and beef meat from a traditional butcher because “they trusted them”. According to data from Ministerio de Agricultura [40], traditional butcher shops were the commercial channel in 2020 that accounted for the highest volume of sheep/goat meat sales, with 39.6% the kilos purchased there, followed by supermarkets and shelf-services.

As shown in Table 6, there were no significant differences among the clusters in the frequency of goat kid meat consumption (χ^2^ > 0.10). Almost all respondents (94.7%) ate goat kid meat less than once per month, while 3.2% never ate this meat and 2.1% consumed it twice or three times per month. In general terms, this meat is not commonly consumed, which accounts for these results. Mandolesi et al. [36] showed that consumers mentioned price as an important quality indicator and perceived lamb and sheep meat as too expensive, and for this reason did not consider it suitable for daily consumption. On the contrary, lamb tends to be more expensive than other meats, and thus lamb meat consumers are less sensitive to variations than meat consumers in general [43]. It is widely known that goat kid meat is more expensive than other meats, but just as with lamb meat, goat meat farms are considered “producers of meat of high functional quality and providers of ecosystem services” [47].

The most frequent place where people ate goat kid meat was a restaurant (64.2%). The “negative” cluster was made up of people who consumed goat kid meat both at home and in a restaurant. Meanwhile, the “idealistic” and “concerned with fat content” clusters were composed of people who consumed it mainly in a restaurant. Lastly, the “positive” cluster was less defined in this regard, since its members were divided between those who consumed goat kid meat at another person’s house and those who did so in a restaurant. Greater neighborhood spatial access to restaurants was associated with a lower frequency of home cooking [48]. The results by Bernués et al. [10] showed that a higher proportion of younger respondents liked eating out and going to restaurants and they also showed a greater liking for changes in meals, although they did not enjoy cooking.

The “negative” cluster was characterized by a tendency (although not with significant differences) to cook at home more frequently than the other clusters, which could be linked to the fact that, as indicated above, they preferred to eat goat kid meat at home. The variable “How often do you eat out each week?” showed significant differences between clusters, with the “concerned with fat content” cluster being the group that did so more frequently, which is maybe linked to their preference to eat goat kid meat in restaurants. Almost 50% of the families never consumed prepared meals and around 40% only did so once or twice per week, although without significant differences between clusters.

### 3.4. Importance of Meat Attributes

Table 7 shows the importance of the meat attributes for the four clusters of consumers. Only those in which there were differences between clusters are shown.

There was a significant difference among clusters on the importance of animal breeding (χ^2^ < 0.10), although, in general, it was not a factor that was considered very important (around 70% of the respondents positioned it at level 1 or 2 of the scale). The “negative” and “concerned with fat content” clusters were the ones that considered this factor more important compared to the “idealistic and “positive” clusters. Ripoll et al. [31] reported that there was no breed effect on goat kid meat flavor when valued by a trained sensory panel. On the contrary, according to Guerrero et al. (2014) [49], consumers perceived differences in the quality of the meat from various breeds of suckling kids, although differences were not statistically significant when suckling kids were compared to lambs that had quality certification (PGI). The results found by Sañudo et al. [41] did not show a clear relationship between sheep breed or sex and flavor when evaluated by consumers from six European countries.

There was a significant difference among clusters on the importance of animal live weight (χ^2^ < 0.10) and 61.9% of respondents thought that this factor was important or very important. The frequency of respondents who considered this factor important was higher in the “negative” cluster than in the rest. Only 12% of the respondents answered that the weight of the animals was not an important factor, with the majority of them belonging to the “concerned with fat content” cluster. Ripoll et al. [31] showed a relationship between the weight of the animals and certain parameters evaluated by a trained sensory panel, with light-weight kids being more tender, juicy, and having less kid and milk odors than heavier kids. In the same way, Rodrigues and Teixeira [26] concluded that the consumers’ panel preferred lighter carcasses for all of the sensory variables. Ripoll et al. [31] found an important effect of the breed and live weight in suckling kids on the instrumental parameters of texture.

In addition, there was a significant difference among clusters over the importance of animal feeding (χ^2^ < 0.10). In the “concerned with fat content” cluster, 22.7% of the respondents considered this factor to be of little importance, while in the “idealistic” cluster, 64.7% considered it of intermediate importance; the “negative” cluster was the group that considered it most important, with 46.2% considering it to be very important. Font i Furnols et al. [22] observed that for some types of consumers, the feeding system was the most important factor. A study by Mandolesi et al. [36] revealed that the majority of the respondents specified that the quality of meat “reflects how the animal was fed”.

Regarding these three attributes (breed, live-weight, and feeding), the “negative” cluster is the one that evaluated them as most important, whereas the “concerned with fat content” cluster considered them of less importance, and the “idealistic” and “positive” clusters were considered of intermediate importance.

There was a significant difference among clusters in the importance of meat tenderness (χ^2^ < 0.10), with 86.0% of the respondents considering that it was an important factor to consider. Of these, in the “negative” cluster, 76.9% stated that it was a very interesting factor, while those who thought that it was a factor of medium importance (7.5% of the overall total) were mainly in the “concerned with fat content” cluster. For Jia et al. [50], goat meat tenderness was the main quality index that defines the consumer satisfaction and the intention of repeating the purchase.

There was a significant difference among clusters in the importance of meat taste (χ^2^ < 0.10), with 69.9% of the respondents considering it very important, which rose to 84.6% in the “negative “cluster. These results are in agreement with Ripoll et al. [31] and Verbeke and Vackier [51], who identified a group of consumers labeled “straightforward meat lovers,” who focus mainly on taste as the decisive criterion for meat appraisal. Acceptance of goat kid meat due to its taste is strongly linked to its historical and cultural uniqueness right through the production, marketing, and consumption chains [52]. Similarly, dos Santos Souza et al. [53] specified the negative perceptions of goat meat quality being partially due to the lack of familiarity with the meat, especially in communities that do not produce goat meat, as is the case with the clusters composed mostly of residents from Seville, who were mainly found in the “concerned with fat content” cluster.

For the other attributes we considered, there were no differences between clusters, and they are therefore not shown in Table 7; however, they were considered important by the consumers and deserve a brief mention.

Regarding the type of meat cut, most of the consumers were not concerned about which cut of meat they consumed, so 38% of the respondents considered this factor unimportant. This may be because they already have a lot of information about the relationship between cuts and their suitability for cooking. For Mandolesi et al. [36], small cuts (e.g., chops, ribs, or arrosticini) were preferred, because they were considered easier to prepare and cook, especially among the younger participants.

The place of consumption was also considered an unimportant attribute by 75% of the participants. In the same way, according to Sañudo et al. [54], breed and place of consumption were the factors valued least.

As for meat color, 60.3% of the participants considered it to be a factor of medium importance. Since it is well known that color is a consumer attribute for fresh meat [29], meat color appeared to be a recurring intrinsic factor in assessing consumer preferences.

As regards the animal’s age, 67% of the respondents considered it to be a very important factor. It is well known that an intense flavor is usually associated with older animals, and the cooked meat of older goats is often criticized for having a “goaty odor” [55], whereas meat from younger goats has a more desirable flavor [27].

The origin of the meat was not considered important by any of the consumer groups. Some studies have reported that local breeds of lamb are better valued by consumers in their production area than foreign breeds [10,41,56]. For Bernues et al. [20] and Font i Furnols et al. [22], the origin of the meat was the most important factor in determining consumers’ purchasing decisions. Nevertheless, our results do not agree with this tendency, which could be due to a lack of knowledge about the origin of goat kid meat. As described by Font and Guerrero [57], the meat sector should provide more information and clearer labeling with details about the origin of the meat.

Finally, regarding the amount of fat in the meat, it was observed that 88.2% of the respondents considered it to be very important. Consumers are, in general, concerned about the fat content of meat because they consider the absence of fat to be an indicator of healthiness in meat [58]. Banović et al. [59] highlighted that visible fat content is also a key factor used to assess meat quality; nevertheless, Ripoll et al. [29] reported that meat from light suckling kids contains no visible fat.

### 3.5. Home Test

The results obtained in the consumers’ home-tests are shown in Table 8.

The mean scores given in the evaluation from the sensory analysis, according to the different clusters, are presented in Table 8.

All of the scores given by all clusters were between 6.5 and 8 (on a scale from 1 to 10), indicating that the consumers had, in general, a positive perception of goat kid meat. The “negative” cluster scored the highest (more than 7), being statistically different from the other clusters, among which there were slight or no differences. The highest scores given by all of the consumers for taste and tenderness agree with the previous perception of the importance of these two factors. Suckling kid meat was evaluated very highly by consumers, who considered it to be a quality product when they tasted it. Different studies have demonstrated that suckling kids (young animals that have been only or mainly reared with milk until reaching slaughter weight) provide a high-quality product [28,29,31,60]. The four clusters can be therefore defined as follows:

“Negative” cluster: People who did not have a good perception of goat kid meat; older people, with more established habits of eating certain types of meat. They were concerned about animal live weight and feeding, and attributed the greatest importance to meat tenderness and taste and, coherently, assigned the highest scores in the tasting test for both tenderness and taste.

“Idealistic” cluster: This group attributed relative importance to animal live weight and feeding and considered that taste was more important than tenderness. Accordingly, they tended to assign the lowest scores for taste, but also to tenderness and overall appraisal, perhaps because their own idealization made them expect more from the meat.

“Positive” cluster: This group considered both animal live weight and feeding important and, like the “idealistic” cluster, considered taste more important than tenderness, although the difference was less relevant. Consequently, they assigned medium-high scores for taste.

“Concerned about fat content” cluster: This group resembled the “positive” cluster for the importance of animal live weight and feeding, but they considered taste just as important as tenderness. This cluster was composed of people living in Seville, an area with a very low consumption of lamb or goat kid meat, and this lack of exposure could account for the medium–low scores assigned to the meat.

It is important to note that tenderness was the most outstanding parameter for older people (“negative” cluster), whereas for the other three clusters, composed of younger people, there was not so much difference between the appraisal of all parameters. This fact has been highlighted by other authors [61].

In the current situation of consumption, it is essential to develop practical strategies for increasing goat kid meat consumption. There are certain parameters to consider for the population in general (for example, the weight and feeding of the animals or the amount of fat the meat contains), and the information that consumers receive about the meat quality parameters (for example, its origin) must be improved. Labeling must be improved to achieve this objective, and one research goal for the industry could be the development of an objective grading system to demonstrate eating quality attributes [62].

The main practical implication can be seen from our results: To improve the market for this type of meat, it is important to improve the information offered to consumers. There are three principal reasons for this:(1)There is great potential for creating markets for a variety of goat products in southern European countries [2], similar to the markets for suckling lambs. According to Pophiwa et al. [63], there is a need for breeds that are well adapted to the environment and resistant to diseases, with a potential for quality meat production. Goat production systems should be oriented toward the achievement of a more sustainable product and guaranteeing animal welfare [57]. However, in Spain, there are six Protected Geographical Indications (PGIs) for lamb meat, but only one for goat kid meat.(2)Young consumers should be given more information, trying to capture their interest in the importance of this meat in relation with environmental, social, and animal rights issues and, particularly, health concerns [45].(3)Finally, industries should develop marketing strategies according to the different groups of consumers, mainly depending on the region and habits of consumption. In other words, the marketing mix of variables should be defined depending on the specific market niche, and new ways should be devised to attract consumers (designing new products, price variations, new messages and communication channels, new distribution channels, and so on). Moreover, the marketing mix must be adapted to each product and each consumer [19].

## 4. Conclusions

Consumers were classified into four groups according to their perception of goat kid meat. Those clusters were labeled as “positive,” “negative,” “idealistic,” and “concerned with fat content”. Belonging to these groups affected consumers’ sensory appraisal of the meat. The animal’s breed, the way the animal was fed, the type of meat cut, the place of consumption, and the origin of the meat were not considered as decisive factors by most respondents, whereas the animal’s live weight, age, meat tenderness, taste, and amount of fat were considered very important. Color was described as an attribute of medium importance in relation to kid goat meat. Considering that in general, there is widespread ignorance about the factors affecting meat production systems and meat quality, and bearing in mind the differences in the beliefs and attitudes among the four clusters, specific strategies for marketing goat kid meat should be implemented to target the different kinds of consumers.

## Figures and Tables

**Table 1 animals-13-02383-t001:** Questionnaire on the consumers’ sociodemographic variables, purchasing and eating habits, and importance of extrinsic and extrinsic meat attributes.

Socio-Demographic Variables	
Family members ^1^	Under 14 years old
	From 14–65 years old
	Over 65 years old
Gender ^2^	Male
	Female
Age ^1^	
Place of residence	Zaragoza
	Sevilla
**Purchasing and eating habits**	
Where do you buy meat more often?	Traditional butcher
	Local market
	Supermarket
	Hypermarket
How often do you eat kid meat?	Less than once a month
	2–3 times a month
	1–3 times a week
	More than 3 times a week
Where do you usually eat kid meat?	Own home
	Other people’s homes
	Restaurant
How often do you cook each week? ^3^	
How often do you eat out each week? ^3^	
How often do you eat prepared meals each week? ^3^	
**Questions about the perception of kid meat** ^4^	
Kid meat is more expensive than lamb meat	
Kid meat is healthier than lamb meat	
Kid meat is taster than lamb meat	
Kid meat contains more fat than lamb meat	
I prefer kids being fed by natural milking rather than artificial milking	
I prefer heavier kids to light kids	
I would like to eat more kid meat than I consume now	
I don’t eat more kid meat because I don’t see it in the supermarket	
I would like to buy kid meat from a quality brand	
I would pay more for kid meat from a quality brand	
**Importance of meat attributes** ^5^	
Animal breed	
Animal live weight	
Animal age	
Animal feeding	
Origin of the meat	
Type of meat cut	
Place in which meat is consumed	
Meat color	
Meat tenderness	
Taste of meat	
Amount of fat in the meat	

^1^ Open-ended question; ^2^ dichotomic answer; ^3^ from 0 to 14 (two meals/seven days); ^4^ seven-point Likert scale from “totally disagree” (1) to “totally agree” (7); ^5^ five-point Likert scale from “not important” (1) to “very important” (5).

**Table 2 animals-13-02383-t002:** Sociodemographic variables of the sample.

		Percentage
Age (years)	Under 21	18.9
	From 21 to 30	11.3
	From 31 to 40	11.5
	From 41 to 50	25.7
	From 51 to 60	21.1
	Over 60	11.5
Gender	Male	45.3
	Female	54.7
City	Sevilla	45.3
	Zaragoza	54.7

**Table 3 animals-13-02383-t003:** Consumer groups in terms of the sociodemographic variables.

	CL1 (13.8%)	CL2 (34.0%)	CL3 (27.7%)	CL4 (24.5%)	s.e.	*p*
Age	56 a	38 b	44 b	36 b	4.154	0.006
N° people <14 years at home	0.1 b	0.8 a	0.5 ab	0.5 ab	0.152	0.039
N° people 14–65 years at home	2.8	2.8	3.0	3.0	0.172	0.839
N° people >65 years at home	0.8	0.4	0.4	0.4	0.126	0.207

Different letters mean significant differences (*p* < 0.05).

**Table 4 animals-13-02383-t004:** Consumer groups in terms of place of residence *.

	Total (%)	CL1 (13.8%)	CL2 (34.0%)	CL3 (27.7%)	CL4 (24.5%)	χ^2^
Seville	45.7	11.6	30.3	18.6	39.5	0.016
Zaragoza	54.3	15.7	37.3	35.3	11.7	

* Dichotomic answer.

**Table 5 animals-13-02383-t005:** Perception variables used by the cluster consumers. Means by cluster, standard error, and level of significance in the GLM procedure *.

	CL1 (13.8%)	CL2 (34.0%)	CL3 (27.7%)	CL4 (24.5%)	s.e.	*p*
Kid meat is more expensive than lamb meat.	3.1 c	4.8 ab	5.3 a	4.0 ab	0.258	0.000
Kid meat is healthier than lamb meat.	2.6 c	3.8 b	5.4 a	3.8 b	0.202	0.000
Kid meat is tastier than lamb meat.	2.3 c	4.7 a	5.2 a	4.0 b	0.234	0.000
Kid meat contains more fat than lamb meat.	2.3 b	3.0 ab	2.8 b	3.7 a	0.243	0.004
I prefer kids being fed by natural milking rather than artificial milking.	2.1 c	4.2 a	2.0 c	3.5 b	0.222	0.000
I prefer meat from heavier kids to that of light kids.	1.4 b	1.7 b	1.5 b	4.4 a	0.258	0.000
I would like to eat more kid meat than I consume now.	4.92	5.50	5.43	4.93	0.311	0.202
I do not eat more kid meat because I do not see it in the supermarket.	4.67	5.60	5.26	5.20	0.339	0.873
I would like to eat kid meat from a quality brand.	5.33	5.35	5.57	4.93	0.364	0.380
I would pay more for kid meat from a quality brand.	5.00	4.95	5.39	4.47	0.344	0.267

* A 7-point Likert scale was used (from “totally disagree” to “totally agree”). CL1: “negative”; CL2: “idealistic”; CL3: “positive”; CL4: “concerned with fat content”. Different letters mean significant differences (*p* < 0.05).

**Table 6 animals-13-02383-t006:** Purchasing and eating habits.

		Total (%)	CL1 (13.8%)	CL2 (34.0%)	CL3 (27.7%)	CL4 (24.5%)	χ^2^	
No continuous variables								
How often do you eat kid meat?	I never eat kid meat	3.2	0.0	3.1	0.0	8.7	0.277	
	Less than once a month	94.7	100	90.6	100	91.3		
	2–3 times a month	2.1	0.0	6.3	0.0	0.0		
	More than once a week	0.0	0.0	0.0	0.0	0.0		
Where do you usually eat kid meat?	Own home	19.8	41.7	23.1	12.5	10.6	0.029	
	A different house	16.0	16.7	3.8	33.3	10.5		
	Restaurant	64.2	41.6	73.1	54.2	78.9		
Continuous variables								
			CL1 (13.8%)	CL2 (34.0%)	CL3 (27.7%)	CL4 (24.5%)	s.e.	*p*
How often do you cook each week? (0–14)			6.3	4.4	4.7	4.3	1.02	0.601
How often do you eat out each week? (0–14)			1.5ab	1.4ab	1.2b	2.6a	0.38	0.045
How often do you eat prepared meals each week? (0–14)			0.5	0.9	0.8	1.2	0.24	0.423

CL1: “negative”; CL2: “idealistic”; CL3: “positive”; CL4: “concerned with fat content”. Different letters mean significant differences (*p* < 0.05).

**Table 7 animals-13-02383-t007:** Importance of meat attributes. Only attributes for which statistical differences between clusters were found are shown.

	Level *	Total (%)	CL1 (13.8%)	CL2 (34.0%)	CL3 (27.7%)	CL4 (24.5%)	χ^2^
Animal breed	1	47.8	38.5	58.1	42.3	45.5	0.018
	2	23.9	23.1	25.8	30.8	13.6	
	3	20.7	23.1	16.1	23.1	22.7	
	4	5.4	0	0	3.8	18.2	
	5	2.2	15.4	0	0	0	
Animal live weight	1	12.0	0	3.2	11.5	31.8	0.007
	2	5.4	0	9.7	7.7	0	
	3	20.7	15.4	35.5	19.2	4.5	
	4	35.9	30.8	25.8	46.2	40.9	
	5	26.0	53.8	25.8	15.4	22.7	
Animal feeding	1	10.8	0	9.4	7.7	22.7	0.085
	2	14.0	15.4	15.6	15.4	9.1	
	3	18.3	15.4	34.4	7.7	9.1	
	4	33.2	23.1	31.3	38.5	36.4	
	5	23.7	46.2	9.4	30.8	22.7	
Meat tenderness	1	1.0	0	3.1	0	0	0.099
	2	5.4	0	3.1	11.5	4.5	
	3	7.5	0	0	7.7	22.7	
	4	30.1	23.1	40.6	30.8	18.2	
	5	55.9	76.9	53.1	50.0	54.5	
Meat taste	1	1.0	0	3.1	0	0	0.029
	2	1.1	0	0	3.8	0	
	3	5.4	0	0	0	22.7	
	4	22.6	15.4	21.9	30.8	18.2	
	5	69.9	84.6	75.0	65.4	59.1	

* Five-point Likert Scale from 1 (“not important”) to 5 (“very important”). CL1: “negative”; CL2: “idealistic”; CL3: “positive”; CL4: “concerned with fat content”.

**Table 8 animals-13-02383-t008:** Scores for sensory variables measured in consumers’ home-test, according to clusters established by consumers’ perception about goat kid meat.

	CL1	CL2	CL3	CL4	s.e.	*p*
Taste	8.0 a	6.7 c	7.3 b	6.7 c	0.12	<0.0001
Tenderness	7.3 a	6.5 b	6.9 b	6.7 b	0.15	0.002
Juiciness	7.3 a	6.5 b	6.9 b	6.7 b	0.14	0.005
Overall appraisal	7.9 a	6.9 c	7.3 b	7.0 bc	0.12	<0.0001

CL1: “negative”; CL2: “idealistic”; CL3: “positive”; CL4: “concerned with fat content”. Different letters mean significant differences (*p* < 0.05).

## Data Availability

Not applicable.

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
