# Peer review of "Relationship between Consumers’ Perceptions about Goat Kid Meat and Meat Sensory Appraisal"

_animals, 2023, doi:10.3390/ani13142383_

Round 1
Reviewer 1 Report
The objectives of the study were very clear. The main objective of the study was to investigate consumers' perceptions of children's meat and to examine whether their perceptions were related to their sensory evaluations, which were measured by the average of consumers' household tests. However, the manuscript's extensive use of passive voice and repetitive words lacks professionalism. The whole manuscript needed a lot of grammatical editing. Furthermore, all my suggestions are listed below.
Point 1: Line 38: The manuscript primarily focuses on the discussion of ‘goat kid meat’ rather than ‘goat meat’. Therefore, it is recommended to revise the keywords accordingly.
Point 2: Line 49-52: The ethical implications of such consumption practices, wherein animals are exclusively fed on milk and slaughtered at a live weight of 8-10 kg within 35-45 days, remain debatable.
Point 3: Line 56-58: Is the distinction between red meat and white meat based on their fat content or rather on the amount of myoglobin present in each type?
Point 4: Line 71-73: The manuscript employs an excessive amount of passive voice and recommends sentence restructuring.
Point 5: Line 83-84: The vocabulary is often utilized and it is recommended to replace it with more specialized words such as 'very'.
Point 6: Line 101: The preposition used with the noun must align with its semantics, indicating that "about" should be replaced by "of". I recommend substituting the term "kid meat" with "goat kid meat" to eliminate any potential ambiguity.
Point 7: Line 134-135: Is this method of sensory evaluation justifiable? Are there any credible references available for consultation?
Point 8: Lines 164-167: How do we ensure that each family meets the test requirements?
point 9:Line 531: References are out of date and should be replaced with more recent research.
Moderate editing of English language is required
Reviewer 2 Report
The manuscript entitled by Alcalde et al. "Relationships between consumers’ perception about goat kid meat and meat sensory appraisal" which deal with the correlation between the perception of consumers about goat kid meat and sensorial attributes of such meat. I recommend to take into consideration the following comments:
Title: used word "Correlation" instead of "Relationships"
abstract: improve to be more representative of study
Introduction:
Lines 59: mention full phrase of abbreviation MAPA and the citation
Martial and methods:
Lines 152-155: mention such characteristics (number of animal for each breeds, production system, carcass characteristics, feed) even not in very detailed.
Line 160: you indicated " Each family received weekly one leg ….." how many weeks? Two ?
Line 160-161: you indicated " At the end of the trial, all the families had tasted two legs of each of the seven breeds involved" so 30 families x 7 breed = 210 animals, not 200 animals as you mentioned in line 152, how you explain that?
Line 164-166: mention more details about heat treatment such as heating temperature and the period of treatment
Results and Discussion:
Line 196-198: Basing to what four clusters were created (CL1-CL4)? if was basing on the analysis of table no. 5, so it is better to depend the termes “negative”, “idealistic”, “positive” and “concerned about fat content” instead of CL1, CL2, CL3 and CL4, respectively, in all the work (as in the paragraph of abstract). And so, in this case the table number 5 must be submitted before tables 3 and 4, to avoid the confusion of logical sequence.
Line 207-208: you indicated "The clusters differed in the average age of the respondent, the age of the family members and place of residence (Table 3)", but "place of residence" not mentioned in table 3, actually in table 4.
Line 370-371: correct the sentence
Line 412: you indicated "….. small cuts (e.g., chops, ribs, shish kebab, köfte or arrosticini) were preferred", but "place of residence…", can you explain the location of cuts of shish kebab and köfte in the carcase of animal? as I know that shish kebab and köfte are kinds of foods not cuts of carcases!
Line 474-476: you indicated "It is important to note that tenderness was more important than taste for older people (Cluster 1), whereas it was the other way round for the other three clusters, composed of younger people", this information not completely match with the results in table No. 8.
Conclusions:
Lines 505-510: improve to be more representative of study
Author Response
Attached you will find the document with the responses to the suggested changes.

Round 2
Reviewer 2 Report
The authors take into consideration may recommendations not completely and yet The manuscript needs major modifications:
abstract: yet not improved to be more representative of study
Martial and methods:
Lines 152-155: mention such characteristics (number of animal for each breeds, production system, carcass characteristics, feed) even not in very detailed.
Not are part of the survey (Table 1), I mean (Detailed information about the production system, carcass characteristics and instrumental and sensory meat quality of the animals involved in this study can be seen in Panea, Ripoll, Horcada, Sañudo, Teixeira and Alcalde [5] and Ripoll, et al. [32])
Line 164-166: mention more details about heat treatment such as heating temperature and the period of treatment
It was mandatory for each family to cook all their legs in the same way, but no further details on temperature/time were given to the families. This was specified in the methodology section. The team has extensive experience in developing sensory analysis and took into account that the most important thing was that the families cooked it in the same way, regardless of whether all the families cooked it in the same conditions or in different conditions.
That meaning the all process of sensorial evaluation was not under control of authors and this is a weak point of this study
Author Response
The reviewer's recommendations are answered in the attached file.

Round 3
Reviewer 2 Report
Thank you for your explanation, but still i believe that different temperature/time for meat treatment have high impact on the sensorial properties and espationally on tenderness